# Effects of Black Cumin Seed (*Nigella sativa*) and Coconut Meals (*Cocos nucifera*) on Broiler Performance and Cecal Microbiota

**DOI:** 10.3390/ani13030535

**Published:** 2023-02-02

**Authors:** Ahmed Zaazaa, Samer Mudalal, Mohammed Sabbah, Mohammad Altamimi, Abdelhafeed Dalab, Maen Samara

**Affiliations:** 1Department of Animal Production and Animal Health, Faculty of Agriculture and Veterinary Medicine, An-Najah National University, Nablus P.O. Box 7, Palestine; 2Department of Nutrition and Food Technology, Faculty of Agriculture and Veterinary Medicine, An-Najah National University, Nablus P.O. Box 7, Palestine; 3Department of Veterinary Medicine, Faculty of Agriculture and Veterinary Medicine, An-Najah National University, Nablus P.O. Box 7, Palestine

**Keywords:** black cumin seed meal, coconut meal, broiler performance, cecal microbiota

## Abstract

**Simple Summary:**

Black cumin meal (BCSM) and copra or coconut meal (CM) are by-products resulting from the extraction of black cumin and coconut oil, respectively. They contain valuable functional nutrients such as proteins, dietary fibers and phytochemicals. The objective of this study was to investigate the effects of different levels of BCSM and CM in broiler diets on the growth performance and cecal microbiota of birds. It was found that the addition of coconut meal to broiler diet significantly improved body weight and feed conversion ratio (FCR). In addition, the hot carcass dressing percentage was increased due to the addition of BCSM and CM either separately or in combination compared to the control group. In conclusion, the use of BCSM and CM in broiler diets had positive effects on growth performance and gut health (inhibitory effect against pathogenic microbes and improvement of gut microbiota diversity).

**Abstract:**

The objective of this study was to investigate the effect of dietary supplementation with black cumin seed meal (BCSM) and coconut meal (CM) on the performance and cecal microbiota of Cobb 500 hybrid broilers. The study was conducted on 600 chicks on the first day of hatching; the chicks were randomly distributed equally into 12 equal-sized floor pens. Four dietary treatments (C, T1, T2 and T3) were replicated three times (50 chicks/replicate): C was the control group; T1 was supplemented with 10% BCSM; T2 was supplemented with 10% CM; T3 was supplemented with 5% BCSM and 5% CM. At slaughter age on day 35, our findings showed that treatment T2 increased significantly body weight and feed conversion ratio (FCR) compared to C, T1 and T3. In addition, the hot carcass dressing percentages in treatments T1, T2, and T3 were significantly higher than that of the C group. The results of relative normalized comparative gene expression of *Clostridioides difficile*, *Roseburia* and *Streptococcus* were not significantly changed in all treatments (*p* > 0.05). Treatment T1 resulted in a significant decrease in gene expression of the entire microbiota, while treatment T2 resulted in a significant increase in gene expression of all microbes, leading to an enriched and diverse microbial community. It can be concluded that supplementation with 10% BCSM is beneficial in inhibiting pathogenic microbes during early post-hatch days. In contrast, CM may promote and enhance the diversity of microbial communities during broiler growth. The inclusion of non-conventional feed ingredients in poultry diets may improve growth performance and may reduce the cost of broiler feed.

## 1. Introduction

Black cumin seed (*Nigella Sativa*) meal (BCSM) is a by-product of black cumin oil production. This meal represents a valuable source of plant protein in many countries. It was found that the inclusion of BCSM and cumin seed by-products in broiler diets improves the growth, health and meat characteristics of broilers [1,2,3,4]. Several previous studies have documented the beneficial effects of black cumin seed [4,5,6,7] and black cumin seed oil [8,9,10] on broiler performance traits. The authors of these studies demonstrated that the addition of black cumin seeds at a dosage of 0.5 to 5% improved the growth and health status of broilers. A limited number of studies have been conducted to evaluate the effects of BCSM on broiler growth traits [11,12,13]. El-Deek et al. [11] found that the replacement of soybean meal with BCSM at levels ranging from 0 to 50% had no effect on growth, feed intake, feed conversion ratio and meat characteristics of broilers. Jahan and Khairunnesa [12] found that the inclusion of BCSM at a level of 0.5, 1 or 1.5% in broiler diets had no significant effect on body weight, FCR or meat characteristics compared to broilers on a control diet.

Copra meal or coconut (*Cocos nucifera)* meal (CM) is a by-product obtained from the mechanical extraction of oil from coconuts [14]. The meal obtained from a copra cake is usually pelletized and then used as a feed additive. The nutritional value of CM is different from that of other oilseed meals because it is obtained by mechanical pressing or extraction. It has an oil content of 5 to 15% and 20–25% crude fiber on a dry matter basis [15]. CM is widely used as a feed additive in the rations of ruminants. However, its use in rations for non-ruminants has been limited due to its high fiber content, high content of non-starch polysaccharides, and low contents of lysine, methionine, and cysteine [16].

There were no agreements between previous studies about the effect of CM inclusion in poultry rations on growth performance. Broilers fed diets containing 10–20% CM showed lower growth performance [17,18]. Other researchers reported low feed intake and high water consumption in broilers fed diets containing more than 10% CM [19,20]. On the other hand, it was found that the inclusion of CM (less than 50%) in broiler diets had negative effects on the growth of young birds when compared to the growth of older birds [18,21,22]. It was found that the addition of enzymes (mannanases for instance) can mitigate the negative effects of the fiber content of CM on broiler growth performance [23]. The inclusion of CM in broiler diets had no negative effects on the carcass characteristics of broilers [21,22].

It is well known that broilers are monogastric animals that derive no growth performance benefits from consuming fiber components. Accordingly, these materials were rarely included in broiler diets [24]. However, the physiological functions derived from certain dietary fibers are more valuable than a slight reduction in weight gain. Both broiler producers and consumers are interested in antibiotic-free diets. In this context, improving the gut health microbiota with various functional ingredients such as essential oils from *Origanum* [25] and other extracts from medicinal plants [26,27] may help reduce the use of antibiotics in disease treatment. Antibiotic feed additives such as bacitracin, which is used to prevent enteritis, will cause a significant impact on intestinal microbial communities [28,29]. Dietary fiber intake (as prebiotics) has been found to promote probiotics such as *lactobacilli* and *bifidobacteria* spp. in the intestine which directly compete with pathogens for nutrients or inhibit their growth by producing active molecules [28]. Beneficial bacteria may contribute to energy harvesting in the body and produce short-chain fatty acids which were estimated to provide 10% of the energy of the digestion process [29]. Firmicutes, Bacteroides and Proteobacteria are the most common phyla in the chicken ceca, and sequencing studies have shown *Clostridiales* to be the major member. Some bacteria, although represented in low numbers, are considered opportunistic pathogens, such as *E. coli* [29].

By-products of oilseeds are considered a good source of polysaccharides and potential prebiotics. Additionally, some oilseeds such as black cumin, coriander and sesame seeds are known for their medicinal use. The objective of the current study was to evaluate the effects of different levels of BCSM and CM on growth traits, meat characteristics and microbiota of broiler chicks.

## 2. Materials and Methods

### 2.1. Experimental Design

The experiment was conducted at the farm of An-Najah National University in Tulkarm, Palestine. A total of 600 one-day-old Cobb broiler chicks were purchased from a local hatchery (Palestine Poultry Company, Tulkarm, Palestine). These chicks were randomly housed in an open-sided broiler house that was divided into 12 equal-sized floor pens. Four dietary treatments were replicated three times (50 chicks/replicate). Dietary treatments were as follows: birds in the first treatment (control group) received conventional broiler diets (C); birds in the second treatment (T1) received diets containing 10% BCSM; birds in the third treatment (T2) received diets containing 10% CM; birds in the fourth treatment (T3) received diets containing 5% BCSM and 5% CM. Birds were housed according to the Cobb broiler management guide.

Chicks were raised on wood shavings 10 cm deep. Moisture in the litter was managed daily; wet spots were removed and new shavings were added whenever necessary. The house temperature was 32 °C for the first seven days and was lowered by 2.5 °C weekly thereafter. Chicks were exposed to a conventional light regimen; birds were exposed to 24 h of light for the first 4 days and 23 h of light and 1 h of darkness thereafter.

Iso-energetic and iso-protein diets were formulated in mash form in accordance with the recommended [30] nutrient requirements for broilers. Two types of rations for each dietary treatment (Table 1 and Table 2) were formulated and were given on an ad libitum basis: a starter ration that was given to chicks from day 1 to day 21 and a grower ration that was given to chicks from day 22 to day 35. The added quantities of BCSM and CM were replaced by some ingredients listed in the control diet at levels listed in Table 1 and Table 2. Feed ingredients were obtained from the Palestine Poultry Company. All diets were mixed using a conventional cement mixer. Body weight and feed intake for all birds were recorded weekly. Birds in every dietary treatment were weighed on the last day of every week at the same time of the day. Mortality was monitored daily.

### 2.2. Performance Traits

Body weight gain was calculated by subtracting the live weight at the beginning of the week from the live body weight at the end of the week. Average feed consumed per bird was calculated by dividing the amount of consumed feed by the number of chicks of every dietary treatment. FCR was calculated weekly as the amount of feed consumption per average of body weight gain (average weekly feed consumption (g)/average weekly gain (g)).

At the end of the experimental period, nine chicks were randomly selected from each replicate. The birds were slaughtered to determine meat yield and characteristics of cut-up parts. The selected birds were slaughtered and processed in a small-scale facility. After slaughter, birds were bled, scalded (60 °C for one minute), plucked, eviscerated, dressed and finally dissected according to the commercial protocols of the facility. The initial body weight of each bird was recorded. The carcasses were firstly cooled with tap water at ambient temperature, followed by cooling in a refrigerator at 4 °C for the next day. Carcass, plucked and dressed weights (Dressing % = Carcass weight/live weight * 100) were then measured. Following cutting, the breasts, thighs, drumsticks, wings, neck, legs, head and giblets (gizzard, liver and heart) of each slaughtered bird were weighed.

### 2.3. Cecal Microbiota DNA Extraction and Relative Quantitative RT-qPCR

At 35 days of age, euthanasia was performed humanely by manual cervical dislocation (CD) and confirmed by loss of all reflexes and musculoskeletal movements; luminal contents of the cecal samples were collected by squeezing the ruptured ceca and extracted from five chicks from each group at the end of the experiment at 35 days of age. Total DNA was extracted from 220 mg of cecal luminal contents according to the manufacturer’s protocol of the QIAamp DNA Stool Kit (QIAGEN, Redwood City, California, USA). Gene expression levels of cecal microbiota were examined and evaluated by relative quantitative real-time RT-qPCR analysis. Primers targeting 11 cecal microbiota were used, as shown in Table 3.

### 2.4. Relative Quantification Protocol

Relative quantitative CFX96 Touch Real Time qPCR analysis (BIO-RAD, Hercules, CA, USA) was performed using the GoTaq qPCR Master Mix kit (Promega, Madison, CA, USA). The 20 µL reaction mix was prepared from 10 µL of the GoTaq qPCR Master Mix (2X), 2 µL of the forward primer pm/μL, 2 µL of the reverse primer pm/μL (Table 3), 2 µL of DNA from the sample and 4 µL of nuclease-free water. Cycling parameters were 95 °C for 1 min, 40 cycles at 95 °C for 10 s, followed by 30 s at 60 °C, and 72 °C for 10 s with a final melting at 95 °C for 20 s. Triplicates from each DNA were analyzed, fluorescence emission was detected and relative quantification was calculated automatically according to the internal housekeeping control to normalize the threshold cycle (Ct) values of the other transcripts.

### 2.5. Statistical Analysis

The original data were arranged using Excel 2007 software (Microsoft Corporation, Redmond, WA, USA). Gene expression levels were expressed as means ± SE. The relative quantitative expression results were calculated using the comparative ct-(2^−ΔΔCt^) method according to Livak and Schmittgen [31]. A two-way ANOVA followed by an all-pairs Bonferroni test was applied to compare the means of the treatment groups for different traits using IBM SPSS Statistics 20 software (IBM, Chicago, USA). Differences were considered significant at *p* < 0.05.

## 3. Results

### 3.1. Performance Traits

The effects of dietary treatment on feed intake, body weight and FCR from day 1 to day 35 of the experiment are shown in Table 4. There was no significant difference in cumulative weekly feed intake (in kilograms) between the different dietary treatments.

A comparison of the weekly body weight (on a cumulative basis) of the experimental birds revealed a significant difference. Birds fed the diet containing 10% CM had the highest body weight compared to body weight in treatments C, T1 and T3 at different ages (1–7, 8–14, 15–21, 22–28, and 29–35 days). There were no significant differences in body weights between C, T1 and T3. Differences in FCR between treatments were more significant at ages 22–28 days and 29–35 days than at ages 1–7 days and 8–14 days. Birds in treatment T2 had significantly lower FCR than treatments C, T1 and T3 at 22–28 days and 29–35 days of age. At ages 1–7 days and 8–14 days, treatment T3 exhibited significantly higher FCR than treatment T2 while the remaining treatments had moderate differences.

After completion of the experiment, nine birds from each replicate in each feeding treatment were randomly selected and sacrificed to determine meat yield and percentage of cuts. The effects of BCSM, CM and a combination of both meals on meat yield and cut percentages (expressed as g/whole carcass weight) are shown in Table 5.

Birds in treatment T2 had significantly higher live weight (2109 vs. 1919 g, *p* < 0.05), carcass weight (1646 vs. 1423, *p* < 0.05) and dressing percentage (77.89 vs. 74.63%, *p* < 0.05) than those in control treatment C. There were no differences in live weight, carcass weight and dressing percentage between treatments T1 and T3. Treatment T2 had significantly higher breast and drumstick weights than other treatments. Treatment T3 had higher intestine, liver, gizzard, crop and proventriculus weights than control treatment C, while there were no significant differences between treatments T1 and T2. The inclusion of black cumin seed meal and coconut meal either alone or in combination had no effect on the weights of the thigh, wings, neck, back, legs, head and heart.

### 3.2. Cecal Microbiota

Figure 1 shows the results of gene expression of C, T1, T2 and T3 cecal microbiota of 35-day-old broilers. The microbiota (*Clostridioides difficile*, *Roseburia* and *Streptococcus*) did not reach a significant level in any of the treatments or the control. T1 decreased all other microbiota significantly, causing lower expression of all microbial communities, while T2 increased all the microbial expression levels, causing enriched and diverse microbial communities. The mixture of BCSM and CM in T3 caused a variation in gene expression; *Bifidobacterium* and *Lactobacillus* were downregulated while *Akkermansia muciniphila* and *Butyricicoccus* increased when compared to the control. *Escherichia coli* and *Enterobacteriaceae* were less expressed in treatments than in the control (C; control diet).

## 4. Discussion

### 4.1. Performance Traits

No significant differences in feed intake were observed between dietary treatments throughout the experiment. These results were in agreement with the results of Lee et al. [32]. Our results did not agree with the results of Panigrahi et al. [19] and Sundu et al. [20], who observed that diets containing 10% CM resulted in lower feed intake in broilers. This disagreement may be attributed to the differences in the experimental design (in our study, an open housing system was used), the genotype of the birds and other farming conditions.

The body weights of broilers in the current study were significantly different between dietary treatments, with those fed CM having the highest body weight throughout the experiment. There were no differences in body weight and FCR of broilers fed the control diet or the diets containing BCSM or a combination of both BCSM and CM. The results of the current study were in agreement with those reported by EL-Deek et al. [11] and Sundu et al. [18]. The results of the current study indicated that broilers fed CM had significantly higher body weight and lower FCR compared to those of birds in the other dietary treatments. Our results were inconsistent with those of Sundu et al. [17] and Sundu et al. [18], who observed lowered performance of broilers given diets containing 10–20% CM. EL-Deek et al. [11] reported that replacing soybean with BCSM up to 50% did not affect the growth and meat characteristics of broilers. Broilers given BCSM or a combination of BCSM and CM in the current study had no significant differences with respect to dressing percentage and cut weights. However, birds fed CM had a significantly higher dressing percentage and significantly higher cut weights. Few studies have been conducted to evaluate the effects of CM on the carcass characteristics of broilers. Our results were in agreement with those reported by Jacome et al. [21] and Bastos et al. [22] who observed no negative effects on carcass characteristics of broilers given CM. The differences between the results of previous studies and those of our study may be attributed to the differences in nutritional profile, nutrient availability, and intake pattern of CM and BCSM.

### 4.2. Cecal Microbiota

Coconut is mainly found in tropical and subtropical areas and has several health-promoting effects such as antiseptic, antitumor, bactericidal, antihelminthic, astringent, diuretic, refrigerant, stomachic, vermifuge, antioxidant and vasorelaxant effects [33]. CM has been poorly studied as a potential functional food, but its prebiotic effects are recognized [34]. On a non-dry basis, a quarter of this residue consists of crude polysaccharides that can be utilized by the gut microbiota. Most studies on the prebiotic effects of CM have focused on pure cultures and the utilization of polysaccharides by probiotics. In the current study, CM resulted in a significant increase in all microbiota studied. Probiotics recognized as beneficial microbiota along with other microbiota were increased in the treatment groups when CM was used. CM did not show selectivity for probiotics but increased the biodiversity of the microbial community in the ceca. Microbiota biodiversity is an indicator of microbial balance [35]. This has a positive effect on host health by inhibiting pathogen colonization, improving the utilization of intestinal contents, producing beneficial metabolites and promoting intestinal motility [36]. Body weight and carcass weight changes were significantly higher in CM compared to the other treatments, but FCR was not significant, although it was the lowest between treatments. This may be attributed partly to the microbial diversity in the gut, which provides an additional source of energy from the feed. In contrast, BCSM did not have similar microbial diversity, and therefore average body weight was lower and FCR was higher. In the ceca, BCSM drastically reduced the microbial communities regardless of whether they were beneficial or harmful bacteria, implying that the gut microbiota was less distinct and its contribution to digestion was minimized. It has been previously shown that BCSM contains antimicrobial agents that can inhibit a broad spectrum of bacteria, fungi and protozoa [37,38]. The mixture of both meals showed intermediate results in microbiota and performance levels. Nevertheless, all treatment groups had lower expression levels of harmful bacteria such as *Enterobacteriaceae* and *E. coli* than those found in the group that received the control diet.

## 5. Conclusions

The current study highlights the role of microbiota in broiler production and the use of feed ingredients that enhance microbial balance in the gut. CM showed superiority over BCSM. However, sequential feeding may be suggested to emphasize the role of microbiota, e.g., feeding with BCSM may be employed for the first week of life to create a low-pathogen environment, and then chicks may be fed with CM to enhance microbial diversity.

## Figures and Tables

**Figure 1 animals-13-00535-f001:**
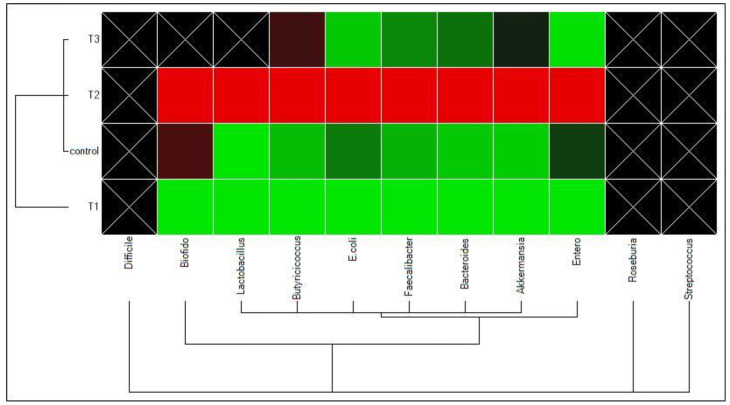
Clustergram of samples and targets in post-hatch chicks (day 35). The data colors indicate greater expression (upregulated—red color), lower expression (downregulated—green color) and no change in expression (no change in regulation—black color). The lighter the color, the greater the expression level determined by CFX96 Touch real-time PCR. C = control diet; T1 = black cumin seed meal 10%; T2 = coconut meal 10%; T3 = 5% black cumin seed meal and 5% coconut meal.

**Table 1 animals-13-00535-t001:** Composition of the starter diets fed to broilers in four feeding trials, g/kg.

Ingredient	C	T1	T2	T3
Yellow corn	317.0	311.5	322.0	236.0
Soybean meal	295.0	292.5	282.0	268.0
Wheat	250	139	139	139
Sunflower	50	50	50	50
Black cumin seed meal	0	100	0	50
Coconut meal	0	0	100	50
Premix ^1^	4	4	4	4
Oil	41	60	60	60
Limestone calcium	13	13	13	13
Dicalcium phosphate	16.5	16.5	16.5	16.5
Sodium bicarbonate	1	1	1	1
Salt	4.5	4.5	4.5	4.5
L-lysine	4.5	4.5	4.5	4.5
DL-methionine	2.5	2.5	2.5	2.5
Threonine	1	1	1	1
Calculated analysis (%)				
Crude protein	22	22	22	22
Crude fat	5.80	6.20	6.40	6.30
Fiber	3.80	3.90	3.90	3.90
Calcium	0.92	0.92	0.92	0.93
Available P	0.47	0.47	0.47	0.47
ME, Kcal/kg ration	3041	3036	3040	3020

Treatments: C = control diet; T1 = black cumin seed meal 10%; T2 = coconut meal 10%; T3 = 5% black cumin seed meal and 5% coconut meal. ^1^ Premix/kg diet: vitamin A, 12,000 IU; vitamin D3, 1500 IU; vitamin E, 50 mg; vitamin K3, 5 mg; vitamin B1, 3 mg; vitamin B2, 6 mg; vitamin B6, 5 mg; vitamin B12, 0.03 mg; niacin, 25 mg; Ca-D-pantothenate, 12 mg; folic acid, 1 mg; D-biotin, 0.05 mg; apo-carotenoic acid ester, 2.5 mg; choline chloride, 400 mg; manganese, 100 g; zinc, 100 g; iron, 40 g; copper, 15 g; iodine, 1 g; cobalt, 0.2 g; selenium, 0.35 g; wheat enzyme, 100 g; phytase, 750 FTU; Lasalocid, 100 g; Bacitracin Methylene Disalicylate (BMD), 55 g. ME: metabolizable energy.

**Table 2 animals-13-00535-t002:** Composition of the grower diets fed to broilers in four feeding trials, g/kg.

Ingredient	C	T1	T2	T3
Yellow corn	479.0	387.0	397.0	329.5
Soybean meal	242.0	229.0	219.0	213.5
Wheat	150	150	150	100
Sunflower	60	60	60	60
Black cumin seed meal	0	100	0	50
Coconut meal	0	0	100	50
Premix ^1^	4	4	4	4
Oil	30	35	35	58
Limestone Calcium	11.2	11.2	11.2	11.2
Dicalcium phosphate	14	14	14	14
Sodium bicarbonate	1	1	1	1
Salt	2.5	2.5	2.5	2.5
L-lysine	4	4	4	4
DL-methionine	1.9	1.9	1.9	1.9
Threonine	0.4	0.4	0.4	0.4
Calculated analysis (%)				
Crude protein	20	20	20	20
Crude fat	4.20	4.40	4.40	6.40
Fiber	3.90	4.10	4.20	4.20
Calcium	0.90	0.90	0.90	0.90
Available P	0.46	0.46	0.46	0.46
ME, Kcal/kg ration	3062	3010	3010	3040

Treatments: C = control diet; T1 = black cumin seed meal 10%; T2 = coconut meal 10%; T3 = 5% black cumin seed meal and 5% coconut meal. ^1^ Premix/kg diet: vitamin A, 12,000 IU, vitamin D3, 1500 IU; vitamin E, 50 mg; vitamin K3, 5 mg; vitamin B1, 3 mg; vitamin B2, 6 mg; vitamin B6, 5 mg; vitamin B12, 0.03 mg; niacin, 25 mg; Ca-D-pantothenate, 12 mg; folic acid, 1 mg; D-biotin, 0.05 mg; apo-carotenoic acid ester, 2.5 mg; choline chloride, 400 mg; manganese, 100 g; zinc, 100 g; iron, 40 g; copper, 15 g; iodine, 1 g; cobalt, 0.2 g; selenium, 0.35 g; wheat enzyme, 100 g; phytase, 750 FTU; Lasalocid, 100 g; Bacitracin Methylene Disalicylate (BMD), 55 g. ME: metabolizable energy.

**Table 3 animals-13-00535-t003:** Real-time qPCR primers sequences targeting different microbiota in the cecal content on post-hatch day 35 chicks.

qPCR (Target Gene)	Pos/Neg (%)	Sequence	Length (bp)
Total bacteria 16S rRNA	96/0	F: ACTCCTACGGGAGGCAGCAGTR: GTAATTCCGCGGCTGCTGGCAC	194–200
*Akkermansia muciniphila*	89/11	F: CAGCACGTGAAGGTGGGGACR: CCTTGCGGTTGGCTTCAGAT	329
*Bacteroides* spp.	100/0	F: GGGTTTAAAGGGAGCGTAGGR: CTACACCACGAATTCCGCCT	116
*Bifidobacterium* spp.	100/0	F: GAATAGCTCCTGGAAACGR: ATAGGACGCGACCCCA	99
*Butyricicoccus* spp.	100/0	F: ACCTGAAGAGAATAAGCTCCR: GATAACGCTTGCTCCCTACGT	69
*Clostridioides difficile*	9/91	F: GCAAGTTGAGCGATTTTACTTCGGTR: GTACTGGCTCACCTTTGATATTYAAGAG	155
*Enterobacteriaceae* spp.	97/0	F: CATTGACGTTACCCGCAGAAGAAGCR: CTCTACGAGACTCAAGCTTGC	190
*Escherichia coli*	97/3	F: CAACGAACTGAACTGGCAGAR: CATTACGCTGCGATGGAT	121
*Faecalibacterium prausnitzii*	100/0	F: GGAGGAAGAAGGTCTTCGGR: AATTCCGCCTACCTCTGCACT	248
*Lactobacillus* spp.	90/9	F: AGCAGTAGGGAATCTTCCAR: CACCGCTACACATGGAG	340–346
*Roseburia* spp.	99/0	F: TACTGCATTGGAAACTGR: CGGCACCGAAGAGCAAT	230
*Streptococcus* spp.	81/12	F: GAAGAATTGCTTGAATTGGTTGAAR: GGACGGTAGTTGTTGAAGAATGG	559

**Table 4 animals-13-00535-t004:** Effects of black cumin seed meal and coconut meal on performance traits of broilers at different rearing intervals.

Traits	Age	C	T1	T2	T3	*p* Value
		M ± SEM	M ± SEM	M ± SEM	M ± SEM	
Cumulative feed intake (g)	1–7 d	132 ± 0.57	131 ± 1.20	130 ± 0.66	131 ± 0.88	0.56
	8–14 d	492 ± 2.00	491 ± 2.40	495 ± 0.33	494 ± 1.16	0.37
	15–21 d	1225 ± 5.78	1215 ± 6.51	1219 ± 0.88	1232 ± 6.35	0.21
	22–28 d	2189 ± 8.57	2177 ± 9.49	2184 ± 2.52	2196 ± 10.17	0.48
	29–35 d	3495 ± 18.27	3471 ± 12.12	3485 ± 0.88	3490 ± 7.53	0.55
Body weight (g)	1–7 d	148 b ± 0.58	148 b ± 0.33	152 a ± 0.58	147 b ± 0.58	<0.05
	8–14 d	450 b ± 1.16	452 b ± 1.20	459 a ± 1.76	450 b ± 0.67	<0.05
	15–21 d	874 b ± 2.40	876 b ± 0.67	885 a ± 1.16	873 b ± 2.60	<0.05
	22–28 d	1464 b ± 2.08	1466 b ± 4.26	1496 a ± 3.18	1465 b ± 4.37	<0.05
	29–35 d	2018 b ± 9.39	2026 b ± 10.11	2097 a ± 10.26	2027 b ± 3.22	<0.05
Cumulative feed conversion ratio (g/g gain)	1–7 d	894 a ± 6.51	887 ab ± 8.09	858 b ± 5.93	890 ab ± 8.54	<0.05
	8–14 d	1092 ab ± 5.21	1086 ab ± 3.06	1080 b ± 4.06	1100 a ± 2.96	<0.05
	15–21 d	1400 ab ± 8.84	1387 ab ± 7.02	1377 b ± 1.20	1412 a ± 10.48	<0.05
	22–28 d	1495 a ± 3.84	1486 a ± 2.19	1457 b ± 1.45	1499 a ± 10.58	<0.05
	29–35 d	1732 a ± 14.85	1714 a ± 3.18	1660 b ± 7.54	1722 a ± 6.39	<0.05

Data are reported as means (M, *n* = 150/group) and standard error of the mean (SEM). Different letters in the same row indicate significant differences (*p* < 0.05). Treatments: C = control diet; T1 = black cumin seed meal 10%; T2 = coconut meal 10%; T3 = 5% black cumin seed meal and 5% coconut meal.

**Table 5 animals-13-00535-t005:** Effect of black cumin seed meal and coconut meal on the relative weight of broiler cuts and organs (g/carcass weight).

Traits	C	T1	T2	T3	*p* Value
	M ± SEM	M ± SEM	M ± SEM	M ± SEM	
Thigh	209 ± 17.54	215 ± 42.73	232 ± 31.76	232 ± 23.61	0.28
Drum sticks	171 ^b^ ± 11.86	191 ^ab^ ± 20.52	201 ^a^ ± 19.48	190 ^ab^ ± 22.33	<0.05
Wings	150 ± 12.99	151 ± 12.17	154 ± 14.75	154 ± 11.09	0.86
Breast	498 ^c^ ± 44.95	579 ^ab^ ± 37.35	632 ^a^ ± 90.31	545 ^bc^ ± 58.97	<0.05
Neck	96 ± 9.19	98 ± 13.76	101 ± 8.14	102 ± 8.87	0.64
Back	199 ± 14.98	200 ± 18.77	211 ± 21.94	202 ± 21.91	0.59
Legs	74 ± 7.02	73 ± 9.29	77 ± 6.74	73 ± 7.78	0.61
Head	33 ± 2.86	35 ± 4.31	37 ± 4.26	37 ± 3.94	0.09
Intestines	99 ^c^ ± 4.71	107 ^b^ ± 7.27	108 ^b^ ± 2.10	124 ^a^ ± 8.58	<0.05
Heart	9.14 ± 0.59	9.95 ± 1.42	10.23 ± 1.55	10.32 ± 1.46	0.23
Liver	41.97 ^b^ ± 3.85	43.41 ^ab^ ± 4.21	45.66 ^ab^ ± 1.42	47.19 ^a^ ± 4.34	<0.05
Gizzard	28.47 ^b^ ± 1.90	29.17 ^ab^ ± 2.74	29.42 ^ab^ ± 2.89	31.9 ^a^ ± 2.62	<0.05
Crop	3.88 ^b^ ± 0.81	3.96 ^b^ ± 0.47	3.98 ^b^ ± 0.61	4.88 ^a^ ± 0.52	<0.05
Proventriculus	7.51 ^b^ ± 0.52	7.74 ^ab^ ± 1.08	8.27 ^ab^ ± 0.44	8.49 ^a^ ± 0.71	<0.05
Live weight	1919 ^b^ ± 49.17	2031 ^ab^ ± 80.87	2109 ^a^ ± 172.95	2030 ^ab^ ± 109.78	<0.05
Carcass weight	1432 ^b^ ± 39.49	1544 ^ab^ ± 106.09	1646 ^a^ ± 187.44	1535 ^ab^ ± 99.99	<0.05
Dressing %	74.63 ^b^ ± 1.14	75.96 ^ab^ ± 3.07	77.89 ^a^ ± 2.55	75.61 ^ab^ ± 1.89	<0.05

Data are reported as means (M, *n* = 27/group) and standard error of the mean (SEM). Different letters in the same row indicate significant differences (*p* < 0.05). Treatments: C = control diet; T1 = black cumin seed meal 10%; T2 = coconut meal 10%; T3 = 5% black cumin seed meal and 5% coconut meal.

## Data Availability

The data presented in this study are available on request from the corresponding author. The data are not publicly available.

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
