# Peer review of "Effects of Black Cumin Seed (*Nigella sativa*) and Coconut Meals (*Cocos nucifera*) on Broiler Performance and Cecal Microbiota"

_animals, 2023, doi:10.3390/ani13030535_

Round 1

Reviewer 1 Report

Lines 15-17: Please change to "Black cumin meal (BCSM) and copra or coconut meal (CM) are byproducts resulting from the extraction of black cumin and coconut oil, respectively."

Line 19: Please change "...Cobb 500 hybrid broilers" to  "...of birds"

Line 19: Please change "feed" to "diet"

Line 20 and Line 33: Please change "percentage of dressing" to "cold/hot carcass percentage"

Lines 38-39: “It can be concluded that supplementation with 10 % BCSM is beneficial to inhibit pathogenic microbes during early post-hatch days.”

How can it be said that pathogenic microbes were inhibited during the early post-hatching period if the samples for microbial analysis were collected on the 35th day?

Lines 44-93: In the introduction, please give some brief information about the microorganisms you identified in the study. Which microorganism is beneficial or harmful?

Line 51: The expression "whole cumin seed" can be confused with Cumin (Cuminum cyminum), please correct it.

Lines 77-78: Please change "Knowing that broilers are monogastric animals that do not benefit from fibrous components in the diet for growth performance." to "It is well known that broilers are monogastric animals that derive no growth performance benefits from feeding fiber components."

Line 80: "overweighing" ??

Lines 81-88: Poor English, please get technical support.

Line 114: If mortality was monitored daily, where did you present the viability results? Readers may also wonder about the effect of feed additives on livability in broilers.
Line 115: The term "parameter" is a technical term belonging to statistics and quantitative genetics. Population statistics are called "parameters". For example, heritability, genetic correlation, etc. are parameters. But live weight or carcass yield is not a parameter, they are phenotypic traits or characteristics. synonymous with characteristic, trait, feature, etc.

Lines 121-122: " At the end of the experiment period, nine chicks from each replicate were randomly selected." Why did you make a paragraph after this sentence? Meaning integrity is broken, there is no continuation.

Lines 126-128: Poor English, please get technical support. "Initial body weight for each bird was recorded, and then weight was recorded prior to every step of processing."
Lines 126-131: Did you cut the carcasses before they had cooled? Please provide additional information. Typically, birds are slaughtered on the first day, and the cold carcass weight is calculated following wet cooling. After waiting 24 hours in a refrigerator, the shredding process is done. Rigor mortis duration and cold chain applications are crucial in terms of meat quality and shelf life in poultry species.

Line 129: drum sticks, ... , legs, ?

Line 132 and Line 140: 
3041 3036 3040 3020 MJ/kg? are you sure? The kilocalories unit number 238.85 kcal converts to 1 MJ.

Line 177: parameters

Lines 176-178: Did you conduct an analysis of variance to compare different characteristics? Or did you use an analysis of variance to compare the means of the treatment groups for a certain characteristic? Please correct.

Table 4: Parameter

Table 4: body weight or gain?

Table 4: cumulative FCR? or weekly FCR?

Table 4: please give in grams instead of kilograms so that the standard errors of the averages can appear.

Lines 205-207: "Despite that scarified birds were randomly selected, birds in treatment T2 had significantly higher live weight (2109 vs. 1919 g, p<0.05), carcass weight (1646 vs. 1423, p<0.05), and dressing percentage (77.89 vs 74.63%, p<0.05) than control treatment C, respectively." Don't you think this is a treatment technical error? Because there are not such big differences between group means in Table 4. Why did you do carcass study on a small number of birds? I wish you had used more animals. For example, there is a difference of almost 150 g even between breast weights.

Lines 222: change “chicks” to “broilers”

Lines 228-229: Which group is it compared to? Please specify.        “Escherichia coli and Enterobacteriaceae were less expressed when compared with the control.”

Line 237: parameter

Lines 238-260: For performance traits, a very weak discussion section has been written; it needs be expanded. It should be stated which changes occur in beneficial and harmful microorganisms and a discussion should be made accordingly. Which microorganisms are considered harmful and which are beneficial should be specified.

Author Response

Dear Reviewer 

I would like to thank you for your efforts in improving our manuscript.

All comments and suggested corrections by the reviewers have been addressed and corrected accordingly. The following text includes our responses to each of the comments and all the corrections have been approved by all authors. In addition, track-changes were used in the manuscript to highlight changes.

Lines 15-17: Please change to "Black cumin meal (BCSM) and copra or coconut meal (CM) are byproducts resulting from the extraction of black cumin and coconut oil, respectively."

It has been changed.

Line 19: Please change "...Cobb 500 hybrid broilers" to  "...of birds"

It has been changed

Line 19: Please change "feed" to "diet"

It has been changed

Line 20 and Line 33: Please change "percentage of dressing" to "cold/hot carcass percentage"

It has been changed

Lines 38-39: “It can be concluded that supplementation with 10 % BCSM is beneficial to inhibit pathogenic microbes during early post-hatch days.”

How can it be said that pathogenic microbes were inhibited during the early post-hatching period if the samples for microbial analysis were collected on the 35th day?

Response: this is an overall conclusion. Based on the impact of BCSM on gut microbiota which has decreased its diversity. BCSM downregulated gene expression of all bacteria including pathogens from day 0 to 35. The post-hatching period is very sensitive in the chick’s life and usually diets are supplemented by antibiotics to inhibit pathogens (Enterobacteriaceae). In this period diverse community of microbes can be detrimental. BCSM can be a safe alternative to antibiotics.

Lines 44-93: In the introduction, please give some brief information about the microorganisms you identified in the study. Which microorganism is beneficial or harmful?

Response: more information was given line 87-93

Line 51: The expression "whole cumin seed" can be confused with Cumin (Cuminum cyminum), please correct it.

It has been corrected

Lines 77-78: Please change "Knowing that broilers are monogastric animals that do not benefit from fibrous components in the diet for growth performance." to "It is well known that broilers are monogastric animals that derive no growth performance benefits from feeding fiber components."

Response: this has been changed.

Line 80: "overweighing" ??

Response: this was changed to give the meaning as” more valuable than’

Lines 81-88: Poor English, please get technical support.

Response: English was revised and corrections were made.

Line 114: If mortality was monitored daily, where did you present the viability results? Readers may also wonder about the effect of feed additives on livability in broilers.

We considered only mortality rate in our study which is commonly used in literature.

Line 115: The term "parameter" is a technical term belonging to statistics and quantitative genetics. Population statistics are called "parameters". For example, heritability, genetic correlation, etc. are parameters. But live weight or carcass yield is not a parameter, they are phenotypic traits or characteristics. synonymous with characteristic, trait, feature, etc.

It has been corrected

Lines 121-122: " At the end of the experiment period, nine chicks from each replicate were randomly selected." Why did you make a paragraph after this sentence? Meaning integrity is broken, there is no continuation.

You are right, it was modified to ensure continuity.

Lines 126-128: Poor English, please get technical support. "Initial body weight for each bird was recorded, and then weight was recorded prior to every step of processing."

It was improved.

Lines 126-131: Did you cut the carcasses before they had cooled? Please provide additional information. Typically, birds are slaughtered on the first day, and the cold carcass weight is calculated following wet cooling. After waiting 24 hours in a refrigerator, the shredding process is done. Rigor mortis duration and cold chain applications are crucial in terms of meat quality and shelf life in poultry species.

Detailed information about cooling process has been added

Line 129: drum sticks, ... , legs, ?

We did not understand this note

Line 132 and Line 140: 3041 3036 3040 3020 MJ/kg? are you sure? The kilocalories unit number 238.85 kcal converts to 1 MJ.

It has been corrected

Line 177: parameters

It has been corrected

Lines 176-178: Did you conduct an analysis of variance to compare different characteristics? Or did you use an analysis of variance to compare the means of the treatment groups for a certain characteristic? Please correct.

You are right. It has been corrected

Table 4: Parameter

It has been corrected

Table 4: body weight or gain?

Body weight

Table 4: cumulative FCR? or weekly FCR?

It has been corrected (cumulative FCR)

Table 4: please give in grams instead of kilograms so that the standard errors of the averages can appear.
It has been changed

Lines 205-207: "Despite that scarified birds were randomly selected, birds in treatment T2 had significantly higher live weight (2109 vs. 1919 g, p<0.05), carcass weight (1646 vs. 1423, p<0.05), and dressing percentage (77.89 vs 74.63%, p<0.05) than control treatment C, respectively." Don't you think this is a treatment technical error? Because there are not such big differences between group means in Table 4. Why did you do carcass study on a small number of birds? I wish you had used more animals. For example, there is a difference of almost 150 g even between breast weights.

You are right the first phrase has been deleted. We think it is not technical error, it is due to treatment effect.

Lines 222: change “chicks” to “broilers”

It has been changed

Lines 228-229: Which group is it compared to? Please specify.        “Escherichia coli and Enterobacteriaceae were less expressed when compared with the control.”

Response: this was clarified. the control (C; control diet).

Line 237: parameter

It has been corrected

Lines 238-260: For performance traits, a very weak discussion section has been written; it needs be expanded. It should be stated which changes occur in beneficial and harmful microorganisms and a discussion should be made accordingly. Which microorganisms are considered harmful and which are beneficial should be specified.

Response: changes in microbiota and which were beneficial or harmful, were added in the discussion lines: 280, 290-291 and 296-297.

Reviewer 2 Report

Introduction

P2, L 77-78: rewrite the sentence

P2, L 79-80: rewrite the sentence

Materials and methods

Please provide data about the approval of Ethics committee for the experiment?

Provide information if the birds were randomly allocated to dietary treatments.

Did the birds have free access to the feed and water? Provide that information

Provide information if the formulated diets were isoprotein and isoenergetic?

Provide more details about the conditions birds were kept during the experiment.

Were the diets provided to the birds in mash or pelleted form?

Results

P6, L 191: delete “(the diet containing 10% CM)”. It is already explained in Materials and Methods

P6, L 192: delete parenthesis

P6, L192-194: there were no significant differences between T2 and T3 at the age 1-7 d

P7, L208: there were no differences

P7, L209-210: along with breast weight, treatment T2 had also higher drum sticks

Discussion

P8, L240-241:  „Diet containg CM or a combination of BCSM and CM decreased feed intake of broilers in the current study“. In the sentence L238-239 you stated there were no significant differences in feed intake between treatments. Also, please check the next sentence.

In the section 4.1 the authors should be focused more on the explaining the obtained results rather that reporting the other findings. This should be notably improved as is lacking in valid discussion.

The discussion in section 4.2 should be also significantly improved.

P9, L264: replace cocos nucifera with coconut

P9, L268-270: support the statements with appropriate references

Uniform the style for reporting the type of the diet (T2, CM or 10%CM).

English language and grammar should be checked throughout the manuscript.

Author Response

I would like to thank you for your efforts in improving our manuscript.

All comments and suggested corrections by the reviewers have been addressed and corrected accordingly. The following text includes our responses to each of the comments and all the corrections have been approved by all authors. In addition, track-changes were used in the manuscript to highlight changes.

Introduction

P2, L 77-78: rewrite the sentence

Response: this has been changed

P2, L 79-80: rewrite the sentence

Response: this has been changed

Materials and methods

Please provide data about the approval of Ethics committee for the experiment?

The data was mentioned under the section of Institutional Review Board Statement

The experiments were conducted at the farm of An-Najah National University. The experiments on the animals were approved by the Palestinian Animal Welfare Committee (protocol code 18C/2021), and care was taken to minimize the number of animals used.

Provide information if the birds were randomly allocated to dietary treatments.

It has been included

Did the birds have free access to the feed and water? Provide that information

It has been included

Provide information if the formulated diets were isoprotein and isoenergetic?

It has been included

Provide more details about the conditions birds were kept during the experiment.

It has been included

Were the diets provided to the birds in mash or pelleted form?

It has been included

Results

P6, L 191: delete “(the diet containing 10% CM)”. It is already explained in Materials and Methods

It has been deleted

P6, L 192: delete parenthesis

It has been deleted

P6, L192-194: there were no significant differences between T2 and T3 at the age 1-7 d

It has been modified

P7, L208: there were no differences

It has been modified

P7, L209-210: along with breast weight, treatment T2 had also higher drum sticks

It has been modified

Discussion

P8, L240-241:  „Diet containg CM or a combination of BCSM and CM decreased feed intake of broilers in the current study“. In the sentence L238-239 you stated there were no significant differences in feed intake between treatments. Also, please check the next sentence.

Yes, you are right, there was a mistake. The paragraph has been corrected

In the section 4.1 the authors should be focused more on the explaining the obtained results rather that reporting the other findings. This should be notably improved as is lacking in valid discussion.

We tried to improve this part

The discussion in section 4.2 should be also significantly improved.

We tried to improve this part

P9, L264: replace cocos nucifera with coconut

It has been changed

P9, L268-270: support the statements with appropriate references

Uniform the style for reporting the type of the diet (T2, CM or 10%CM).

We tried to make consistent style, some time we introduced the results of some studies that used CM but in the same level as we did in our experiment (10%)

English language and grammar should be checked throughout the manuscript.

English language and grammar have been checked

Round 2

Reviewer 1 Report

Dear Authors,

Please check the spelling of Latin and microorganism names.

Please change "parameters" to "traits or characteristics" in the whole article.

Best regards,

Author Response

I would like to thank you for your efforts in improving our manuscript.

All comments and suggested corrections have been addressed and corrected accordingly. The following text includes our responses to each of the comments and all the corrections have been approved by all authors. In addition, track-changes were used in the manuscript to highlight changes.

Please check the spelling of Latin and microorganism names.

The spelling of Latin and microorganism names has been checked and some of them were corrected.

Please change "parameters" to "traits or characteristics" in the whole article.

"Parameters" was replaced by "traits" in the entire article.

Reviewer 2 Report

The revised manuscript (ID: animals-2147732 ) has been significantly improved according to the reviewers' suggestions.

Author Response

I would like to thank your efforts in improving our manuscript.